# A Novel G-Quadruplex Binding Protein in Yeast—Slx9

**DOI:** 10.3390/molecules24091774

**Published:** 2019-05-07

**Authors:** Silvia Götz, Satyaprakash Pandey, Sabrina Bartsch, Stefan Juranek, Katrin Paeschke

**Affiliations:** 1University of Groningen, University Medical Center Groningen, European Research Institute for the Biology of Ageing, 9713 AV Groningen, The Netherlands; 2University of Würzburg, Department of Biochemistry, Biocentre, 97074 Würzburg, Germany; 3University Hospital Bonn, Department of Oncology, Hematology and Rheumatology, 53127 Bonn, Germany; 4Current address: Indian Institute of Technology, Department of Biosciences and Bioengineering, Powai, Mumbai 400076, India

**Keywords:** protein–DNA interaction, *S. cerevisiae*, G-quadruplex formation, genome stability, RecQ helicase

## Abstract

G-quadruplex (G4) structures are highly stable four-stranded DNA and RNA secondary structures held together by non-canonical guanine base pairs. G4 sequence motifs are enriched at specific sites in eukaryotic genomes, suggesting regulatory functions of G4 structures during different biological processes. Considering the high thermodynamic stability of G4 structures, various proteins are necessary for G4 structure formation and unwinding. In a yeast one-hybrid screen, we identified Slx9 as a novel G4-binding protein. We confirmed that Slx9 binds to G4 DNA structures in vitro. Despite these findings, Slx9 binds only insignificantly to G-rich/G4 regions in *Saccharomyces cerevisiae* as demonstrated by genome-wide ChIP-seq analysis. However, Slx9 binding to G4s is significantly increased in the absence of Sgs1, a RecQ helicase that regulates G4 structures. Different genetic and molecular analyses allowed us to propose a model in which Slx9 recognizes and protects stabilized G4 structures in vivo.

## 1. Introduction

The observation that secondary structures within DNA or RNA can influence biological processes had a great impact on modern biology (reviewed in [1,2,3,4]). One prominent example of such a nucleic acid structure is a G-quadruplex (G4). G4 structures are polymorphic and thermodynamic stable structures that can form within DNA and RNA harboring a specific guanine-rich motif (G4 motif) (reviewed in [2,5]).

Although the in vivo existence of G4 structures was controversially discussed in the past, recent results from computational, biochemical, molecular, and genetic studies provided essential data that G4 structures exist in various organisms and can act as a regulatory tool [2,5,6,7,8,9,10].

G4 structures are discussed as being a regulatory tool in the cell for telomere maintenance, transcription, translation, and even origin activation [7,10,11,12,13]. This assumption is further supported by computational and in vitro and in vivo experiments. These data demonstrated that G4 structures form at specific sites (e.g., promoters) and regulate a specific pathway (e.g., transcription) [6,10,14,15,16,17,18] and can thereby become potential therapeutic targets. In contrast to their regulatory role, stable G4 structures can also block biological processes (e.g., replication, transcription) and, in doing so, increase genome instability (reviewed in [2,19]). G4 structures challenge genome stability by interfering with the normal regulatory machinery and/or stalling of cellular processes (e.g., replication). This can result in genomic mutations and deletions [8,20,21,22]. To ensure genome stability, controlled formation and unfolding of G4 structures is essential. Cellular machinery is necessary to regulate the kinetics of formation and unfolding of G4 structures in response to specific stimuli. To this date, multiple helicases, like members of the RecQ (e.g., Sgs1 in yeast or WRN or BLM in human) or Pif1 family and a few other proteins (e.g., CNBP) have been identified to support G4 unfolding [23,24]. Mutation or dysregulation of G4-binding proteins, especially helicases, are implicated in many human diseases [25]. Most helicases exhibit the ability to unwind G4 structures in vitro [26]. Some helicases recognize specific G4 motifs, in vivo, during certain biological processes [23,27]. How these helicases gain specificity is still unclear. A few examples are known where an additional protein supports the function of a given helicase. For example, in yeast, Mms1 supports the binding of Pif1 to a specific subset of G4 targets—those which are located on the lagging strand [28]. Identification of G4-recognizing proteins supports the detection of in vivo-relevant G4 structures. Additionally, the characterization of these proteins indirectly helps to gain insights into G4 function in vivo.

In this study, we used a yeast one-hybrid approach to identify *Saccharomyces cerevisiae* Slx9 as a novel G4-interacting protein. In vitro experiments confirmed the specific binding of Slx9 to G4 structures. Slx9 is a non-essential yeast protein that genetically interacts with the yeast RecQ helicase Sgs1 [29]. Sgs1 is involved in various processes that are linked to genome stability, such as DNA repair and G4 unwinding [27,30,31]. However, the functional relationship between Slx9 and Sgs1 is unclear. To further address Slx9 function at G4 structures in vivo, we mapped Slx9-binding sites, genome-wide, by chromatin immunoprecipitation (ChIP) followed by deep sequencing (ChIP-seq). These analyses revealed that Slx9 does not significantly bind, genome-wide, to G4 motifs. However, in the absence of Sgs1, Slx9 binds robustly to G4 motifs. Similarly, in the presence of hydroxyurea (HU), when more G4 structures are detectable [32], Slx9 binding to G4 motifs is increased. Additional genetic analyses allowed us to propose a mechanistic model addressing how Slx9 recognizes stabilized G4 structures.

## 2. Results

### 2.1. Identification of Slx9 as a Novel G4-Binding Protein in Vivo

G4 structures are evolutionary conserved, dynamic structures involved in various biological processes [2,5,14,17]. To understand the interaction of G4 structures with various proteins in vivo, an unbiased yeast one-hybrid (Y1H) screen was performed. A G4 motif from chromosome IX (G4_IX_) and a mutated version of the same G4 motif (mut-G4_IX_) were cloned upstream of the aureobasidin A resistance gene (*AUR-1C*) (Figure 1A). Mut-G4_IX_ served as a control to remove any non-specific DNA-binding proteins and select for specific G4-binding proteins. A cDNA library comprising of yeast proteins tagged with the galactose activation domain (GAL4-AD) (Dualsystems Biotech) was transformed in the bait strain. An interaction of a protein with the G4_IX_ motif resulted in the expression of the *AUR-1C* gene and, consequently, growth on aureobasidin A medium. In two independent screens, 156 different proteins were identified (SG, KP unpublished data). One of these proteins was Slx9. Slx9 binding to only the G4_IX_ motif, and not to mut-G4_IX_, was observed. Slx9 is a yeast-specific protein that was demonstrated to genetically interact with the yeast RecQ helicase Sgs1 [29]. Most RecQ helicases, including Sgs1, are potent G4 unwinders [26,31], and their function is linked to DNA repair, telomere maintenance, and transcriptional regulation [27,28,29,30,31,32,33,34].

To confirm the interaction between Slx9 and G4 structures, we cloned, expressed, and purified Slx9 from *Escherichia coli* (Figure 1B). Purified protein was subjected to standard filter-binding assays to determine the affinity of Slx9 to G4 structures [35] (Appendix A). Binding of Slx9 to different G4 motifs (G4_IX_, G4_TP_, and G4_rDNA_) and other DNA controls (mut-G4_IX_, dsDNA, forked DNA, and 4-way junctions) was tested. Slx9 showed preferential binding to all tested G4 structures with binding affinity for G4 structures ranging between 210 to 550 nM. Slx9 exhibited binding to the control sequences, but this binding was weaker and never reached 100% (Figure 1C–F, see figure legend for *K_d_* values). The selective binding of Slx9 to G4 structures was confirmed using microscale thermophoresis (MST) analysis (Appendix A). MST is a powerful tool to analyze ligand–molecule interactions [36]. Fluorescently labelled G4_IX_ and mut-G4_IX_ were incubated with Slx9 in increasing concentrations and subjected to MST to quantify the binding affinities. MST analysis confirmed that Slx9 binds G4_IX_ with a better binding affinity than linear G-rich DNA (mut-G4_IX_) (Appendix A). These binding studies further strengthened the Y1H data that Slx9 binds G4 structures.

So far, there are three classes of G4-interacting proteins described in the literature: proteins that support or stabilize G4 formation, proteins that assist their unfolding, and those which recognize formed G4 structures [23,24]. As we detected no unfolding of G4 structures in both binding analyses, we tested whether G4 structures were stabilized by Slx9 binding. G4 structures have a characteristic CD spectrum owing to the π–π interactions between Hoogsteen hydrogen bonds. We assessed the effect of Slx9 on CD spectra of G4_IX_ and mut-G4_IX_. If the G4 structure is stabilized due to Slx9 binding, changes of the characteristic maxima or minima peaks should be detected after addition of the protein. However, Slx9 titration did not result in any observable changes in the signals of G4_IX_ and mut-G4_IX_ spectra, suggesting that the interaction between Slx9 and the G4 motif does not significantly alter the structure and stability of G4 structures (Appendix A).

### 2.2. Slx9 Binds to G-Rich Regions Genome-Wide

To test if in vitro Slx9 binding to G4 structures could be confirmed in vivo, we performed chromatin immunoprecipitation (ChIP) followed by genome-wide sequencing analysis (ChIP-seq) in asynchronous yeast cultures expressing C-terminal Myc-tagged Slx9. Using MACS 2.0, we identified 205 chromosomal binding sites for Slx9 (n = 3) (see Appendix A for peak locations). Peaks were compared to annotated genomic features (centromeres, ARS, promoters), previously identified protein-binding regions (Pif1, γ-H2AX, DNA Pol2), and regions that harbor annotated G4 motifs [14,17,37,38]. Slx9 did not bind significantly to ARS, centromeres, promoters, yH2AX, and Pif1-binding sites. However, we observed a strong correlation of Slx9 peaks to regions with high DNA Pol2 occupancy (*p* = 0.0001) that indicated that the replication fork slows or stalls near Slx9-binding sites. Surprisingly, we did not detect any overlap between annotated G4 motifs and Slx9 peaks. Despite this data, a MEME search resulted in identification of a specific binding motif in 32 of 205 sites (~15.5%) with high G-richness (>60%), greater than the average GC content of the *S. cerevisiae* genome (38%) (Figure 2A). In the past, G4 motifs have been described with a consensus motif G_3_N_1–7_G_3_N_1–7_G_3_N_1–7_G_3_. However, in recent years, alternative, more flexible G4 motifs that can form metastable G4 structures have been described [39,40,41,42]. Slx9-bound regions contained potentially metastable G4 structures with two guanines next to each other (Figure 2A). This indicated that Slx9 did not bind to G4 motifs harboring a consensus sequence in vivo, but can recognize G4 motifs that form flexible, less stable G4 structures. To note, most Slx9-binding sites were detected in non-G-rich regions.

### 2.3. Slx9 Binds in a Sgs1-Dependent Manner to G4 Motifs

It has been discussed, in recent years, that it is unlikely that all G4 structures form simultaneously in a cell. There would be a selective set of G4s that form during specific biological processes, e.g., replication control, under specific conditions (differentiation, stress, starvation) or in the absence of regulatory proteins (e.g., helicases). The RecQ helicase Sgs1 can efficiently unwind G4 structures in vitro [26,30]. Among other non-G4 related defects, G4-dependent transcriptional changes were detected in Sgs1-deficient cells [14,27]. Due to the published genetic interaction between Sgs1 and Slx9, we speculated that without Sgs1, more G4 structures persist in the cell and, consequently, more Slx9 binds to G4 structures. Thus, we monitored Slx9 binding at G4 motifs in the absence of Sgs1. We performed ChIP–qPCR experiments in two different strains. In the first strain, Slx9 was endogenously Myc-tagged in a wild type background (light), and in the second strain, Slx9 was tagged in a *sgs1Δ* (dark) background (Figure 2B). For qPCR analysis, nine different primer pairs were selected to test Slx9 binding: two non-G-rich regions identified as Slx9-binding sites, two G-rich regions (relaxed G4, identified as Slx9-binding sites), three regions harboring a G4 motif (no detectable Slx9 binding by ChIP-seq), and two control regions (no peaks detected by ChIP-seq). Here, and in all subsequent ChIP and qPCR analyses, binding enrichment was plotted as IP values normalized to input values. In concordance with our ChIP-seq results, Slx9 is significantly associated with the four putative Slx9-binding sites (selected from ChIP-seq: non-G-rich, G-rich) in the wild type background, but did not show significant binding to G4 regions or control regions (Figure 2B). On the contrary, in the absence of Sgs1, Slx9 binding was enhanced at regions that could form G4 structures (Figure 2B), whereas binding to the controls and non-G-rich targets was unaffected by the absence of Sgs1. These results indicate that Slx9 binding to G4 structures is enhanced if they are not unfolded by the helicase Sgs1.

### 2.4. Slx9 Recognizes G4 Structures That Are Stabilized In Vivo

To further investigate the connection of Slx9 and Sgs1, we analyzed the doubling time of *slx9Δ, sgs1Δ*, and the double mutant (Figure 3A). Wild type yeast cells have a doubling time of 90 min, as expected. Cells lacking Sgs1 (*sgs1Δ*) or Slx9 (*slx9Δ)* showed minor growth changes in comparison to wild type (94.5 and 97.5 min, respectively). The double mutant (*sgs1Δ slx9Δ*) has a doubling time of 110.9 min, which is slower than any of the single mutants alone (Figure 3A). Although the differences are not huge, they support previously published data indicating that *SLX9* and *SGS1* genetically interact [29]. To test if G4 formation is the cause of these growth defects, we measured the growth of wild type and *slx9Δ* in the presence of G4-stabilizing ligands. *N*-methyl mesoporphyrin IX (NMM) and Phen-DC_3_ are two G4-specific ligands that have been shown to stabilize G4 structures in yeast [14,43]. Yeast cells were incubated with the G4-specific ligand, and the doubling time was calculated. Incubation with either G4 ligand resulted in faster growth in *slx9Δ* in comparison to untreated. The doubling time of *slx9Δ* cells treated with Phen-DC_3_ was 84.8 min, and with NMM 79.2 min, indicating that the deletion of *SLX9* imparts resistance to G4 stabilization (Figure 3B). Notably, the growth of wild type cells did not change upon treatment (Figure 3B). These data are in accordance with previously published screening results using NMM, in which Slx9 was identified as supporting G4 resistance [14].

Slx9 binding to G4 motifs is not detectable in wild type cells, but increased in the absence of Sgs1 (Figure 2B). To test if this enhanced binding is due to G4 structures, we measured Slx9 binding in the presence of Phen-DC_3_ and NMM when the formation of G4 structures is enhanced. For these analyses, the same Slx9-Myc-tagged strains as in the previous experiment were grown in the presence of either 10 µM Phen-DC_3_ or 8 µM NMM. ChIP analysis was performed in an attempt to understand Slx9 binding to G4 and control regions. Both G4-stabilizing ligands caused a 2- to 8-fold increase of Slx9 binding at all tested G4 motifs (Figure 3C). This elevated binding is similar to the binding of Slx9 in the absence of the Sgs1 helicase (Figure 2B). The stronger binding of Slx9 to ligand-stabilized G4 structures supports the claim that Slx9 binds to folded G4 structures.

### 2.5. Slx9 Binding to G4 Structures Affects the “Repair” of These Structures

As a next step, we aimed to identify the process/pathway in which Slx9 acts at folded G4 structures. Due to the link to Sgs1, we hypothesized that Slx9 functions similarly to Sgs1 at G4s. Sgs1 is a multifunctional enzyme with proposed functions during transcription, telomere maintenance, and DNA repair [14,27,30,31]. Cells lacking Sgs1 are HU-sensitive and have transcriptional changes in genes with G4 motifs in their promoters [14,44,45,46]. We analyzed the impact of Slx9 on G4-mediated changes during transcription. We analyzed if *slx9Δ* cells, similar to *sgs1Δ*, show a selective downregulation of mRNAs with G4 motifs in their promoters. To assess this, we monitored changes in mRNA levels of 11 different endogenous loci that harbor G4 motifs and one control region (actin) using qPCR. Five out of 11 regions showed elevated mRNA levels in *slx9Δ* cells compared to wild type (Appendix A). These transcriptional changes indicated that Slx9 affected mRNA levels of a few genes, but these changes did not directly correlate to the presence of G4 motifs in their promoter.

Sgs1 has a role in homologous recombination and *sgs1Δ* cells are sensitive to HU and UV [44,47]. We tested if *slx9Δ* cells are also sensitive to DNA damage agents. Different cell numbers were spotted in a serial dilution on plates with 100 mM HU or were irradiated with 25 J/cm^2^ UV light (254 nm). We found that *slx9Δ* cells did not show any growth defect under any of the tested conditions (Figure 3D–F). As expected, cells lacking Sgs1 (*sgs1Δ*) were sensitive to UV irradiation and HU treatment. Strikingly, the double deletion *slx9Δ sgs1Δ* showed no growth defects in either of the DNA damaging conditions, indicating that deletions of Slx9 rescued the HU and UV sensitivity of *sgs1Δ* cells (Figure 3E,F).

Treatment of yeast cells with HU is tightly correlated with increased genome instability. To test if our experimental observations can be explained by increased genome instability, we measured the direct correlation between G4 motifs and genome stability in *slx9Δ* cells. We performed a previously published gross chromosomal rearrangement assay (GCR), which measures telomere addition, recombination, deletions, and mutations as a result of a specific insert [48,49]. The GCR assay can quantitatively detect complex genome rearrangements by the loss of two counter-selectable markers (*URA3*, *CAN1*). The two markers are located downstream of the *PRB1* locus. If the inserted sequence increases genome instability, the markers are lost, and the cells can grow on selective media. The growth of colonies can be directly used as a quantitative readout for genome instability, and GCR rates can be determined via fluctuation analysis [50]. Wild type cells have a GCR rate of 0.1 × 10^−9^ events per generation [8]. We monitored genome instability using either a non-G-rich, a G-rich or a G4 insert in either wild type, *sgs1Δ*, or *slx9Δ* cells. Depicted GCR rates were normalized to wild type. No changes in GCR rates were detected in any of the experimental conditions, suggesting Slx9 does not have a direct influence on genome stability (Appendix A).

HU increases genome instability by depleting the dNTP pool of the cell, resulting in replication fork progression changes and, consequently, genome instability [51,52]. It has been observed that HU treatment results in elevated G4 structure formation [32]. We speculated that G4 stabilization by HU was the reason why *sgs1Δ* cells were HU-sensitive, and Slx9 recognized these stabilized G4s. We performed ChIP and qPCR experiments using strains with tagged Slx9 in the presence of HU. qPCR analyses using the same primers as used above revealed that Slx9 binding to G4 targets was stimulated in the presence of HU (Figure 3G). These observations further strengthened our previous results and establish a link between Slx9 and Sgs1 under stress conditions.

## 3. Discussion

Different in vitro and in vivo experiments in various organisms demonstrated the regulatory potential of G4 structures, but also highlight the challenges that such a structure poses for the cell [2,8,20,21,26]. Spatiotemporal regulation of the formation and unwinding of G4 structures is necessary for maintaining genome stability. However, it is not fully understood which proteins support G4 structure formation or recognize folded G4 structures in the cell.

In this study, we performed an unbiased Y1H screen to identify G4-interacting proteins. Among the identified putative G4-binding proteins, Slx9 was chosen for further analysis of its role in G4-mediated processes. Slx9 is a yeast-specific protein and has been reported to function during ribosome biogenesis [53]. Filter-binding assays and MST was used to analyze the binding affinity of Slx9 to G4 structures in vitro. Both techniques showed a preferential binding of Slx9 to G4 structures compared to other DNA structures (Figure 1C–F, Appendix A). The specific binding to G4 structures in vitro is interesting, and points towards a similar binding preference in vivo. However, ChIP-seq experiments did not show a significant correlation of Slx9 binding to previously identified G4 motifs (Figure 2) [17]. Recent findings demonstrated that G4 structures could also form within more flexible G4 motifs [39,40,41,42]. These metastable G4 motifs have two guanines in the G-tract and longer loop regions. Although 15.5% of all Slx9-binding sites (Figure 2A) harbor such flexible G4 motifs, the in vivo G4-specific interaction of Slx9 is rather poor. This indicated to us that Slx9 has no or only a minor function at G4 structures in wild type yeast cells. We went on to determine the specific conditions of G4-binding by Slx9. Due to the published connection of Slx9 and Sgs1 [29] and the link of Sgs1 to G4 motifs [27,31], we checked if G4 structures which remain folded in *sgs1Δ* cells are bound by Slx9. Indeed, we observed a 2- to 6-fold enrichment of Slx9 binding to these G4 structures if Sgs1 was not present in the cell (Figure 2B). No significant changes of Slx9 binding to relaxed G4s was determined by this method, indicating that Sgs1 did not target these regions. These data led to the speculation that either Slx9 at G4s is displaced by Sgs1 and, therefore, little or no binding can be observed in wild type, or that Slx9 cannot bind to G4 region because these structures are too efficiently regulated by the cellular machinery and not stably folded enough in wild type conditions.

Since Slx9 did not alter genome stability (as measured by GCR) or cause transcriptional changes in a G4-specific manner (Appendix A) we speculated that Slx9 binds to folded G4 structures. To test this hypothesis, we stabilized G4 structures by adding the G4-stabilization ligand NMM, or Phen-DC_3_, to yeast cells [14,43]. ChIP analysis showed a 2- to 8-fold enrichment in Slx9 binding to G4s, similar to *sgs1Δ* cells (Figure 3C). In addition to helicase depletion and addition of G4-stabilizing ligand, it has been observed that HU favors G4 structure formation [32]. We speculated that Slx9 binding will be enhanced at G4 structures after HU treatment. ChIP analyses of Slx9 binding confirmed that Slx9 binding is enhanced at G4 motifs in the presence of HU (Figure 3G). Interestingly, the binding of Slx9 only increased mildly at relaxed G4 motifs, which might indicate that these relaxed G4s are not affected by NMM or PhenDC_3_, and that their formation is only mildly stimulated under HU conditions.

In the past, it has been demonstrated that folded G4 structures stimulate genome instability [8,19,25,26,54,55,56,57]. A reference for genome stability is cellular fitness or doubling time. We observed that the doubling time of wild type cells was not affected by the addition of a G4-stabilization ligand. However, in the absence of Slx9, cells displayed a faster growth rate than wild type cells if G4s were stabilized by G4-stabilization ligands (Figure 3B). This result was in agreement with previously published screening result using NMM, in which Slx9 was identified to support G4 resistance [14]. In line with our model, we speculated that Slx9 protects folded G4 structures, meaning that Slx9 blocks access to G4. This can be further expanded to HU conditions: in the absence of Sgs1, cells are HU-sensitive. Whether this is due to folded G4 structures or other features is not known [44]. We observed that cells lacking Slx9 and Sgs1 were not HU-sensitive (Figure 3E,F). Taking these results together, we propose that under HU conditions, more G4 structures form, and more Slx9 proteins recognize and bind these folded structures. In the absence of Sgs1, cells are sensitive to HU because Slx9 still binds and recognizes folded G4 structures and prevents other proteins unfolding them. In the double deletion mutant (*slx9Δ sgs1Δ*), similar to the NMM or Phen-DC_3_ conditions, G4 structures are accessible by other proteins and can be regulated.

From this yeast study, we gained two important insights. First, we revealed that Slx9 recognizes and protects folded G4 structures (Figure 4). Second, although a protein is a strong G4 binder in vitro, the in vivo binding might be different. The identification and characterization of the correct binding condition is essential not only for understanding the correct function of the protein, but also to understand the impact and function on G4 structures in the cell, in the case of Slx9. We speculate that more proteins will be identified that recognize and control G4 formation only under specific conditions, because G4 structures are highly dynamic structures that are altered during the cell cycle, different cellular conditions, and during cellular differentiation.

## 4. Materials and Methods

### 4.1. Yeast One-Hybrid Screen

The yeast one-hybrid screen was performed using the Matchmaker^TM^ Gold Yeast One-Hybrid Library Screening (Clontech, Kyoto, Japan). All yeast strains used in this assay are listed in the Appendix A (Appendix A). To construct the screening strain, a *S. cerevisiae* G4 motif from chromosome IX (G4_IX_) with short flanking regions was cloned into the *S. cerevisiae* Y1HGold genome as described in the manual. The control bait, mut-G4_IX_, was cloned using the same strategy. After determination of the minimal inhibitory concentration of aureobasidin A (AbA), screens were performed using an *S. cerevisiae* DUALhybrid cDNA library (Appendix A) and 5 and 10 mM hydroxyurea (HU) in the plates. cDNA library plasmid (7 µg) was transformed into the screening strain bait G4 according to the manufacturer’s protocol.

After streaking out each yeast colony twice on selective plates, the library plasmids were isolated from overnight cultures. Lysis was performed using DNA lysis buffer (2% Triton X-100, 1% SDS, 100 mM NaCl, 10 mM Tris/HCl pH 8.0, 1 mM EDTA) and glass beads in a FastPrep instrument (MP Biomedicals, Santa Ana, CA, USA) for 1 min at 4 °C, followed by phenol/chloroform extraction and ethanol precipitation. Plasmids were transformed in *E. coli* (XL-1 Blue), and overnight cultures were used to isolate plasmids by alkaline lysis. The obtained library plasmid was sent for sequencing using the primer GAL4ADseq (sequence from Dualsystems Biotech): 5′-ACCACTACAATGGATGATG-3′. 

### 4.2. Cloning, Expression, and Purification of Slx9

*SLX9* was amplified by PCR from *S. cerevisiae* genomic DNA and PCR primers SG304 (5′-AAAAAA*gaattc*ATGGTTGCTAAGAAGAGAAACA-3′) and SG305 (5′-AAAAAA*gcggccgc*TCATTGTTTTTGCAGCTTGATAA-3′). *SLX9* was cloned into the *EcoR*I and *Not*I sites of a pET28a vector (Novagen, Darmstadt, Germany). The resulting construct was confirmed by sequencing. 6×His-tagged Slx9 was expressed in Rosetta pLysS cells grown in LB medium supplemented with 25 µg/mL kanamycin and 30 µg/mL chloramphenicol. Expression was induced by 1 mM isopropyl β-d-thiogalactoside (IPTG) at 18 °C overnight, following the manufacturer’s protocol and established protocols [58].

All purification steps were carried out at 4 °C. Cell lysis was performed in lysis buffer (300 mM NaCl, 20 mM HEPES pH 7.5, 10% (*v*/*v*) glycerol, 1 mM DTT, 5 mM imidazole) by sonication (6 × 45 s, 50% pulse intensity) using a Branson sonifier W250-D (Brandson, Danbury, CT, USA). After centrifugation of the lysate (10,000 g, 30 min) the supernatant was applied onto a Ni-NTA agarose column (equilibrated with two column volume lysis buffer) by gravity. After three washing steps with one column volume wash buffer (300 mM NaCl, 20 mM HEPES pH 7.5, 10% (v/v) glycerol, 1 mM DTT, 15 mM imidazole), the bound protein was eluted with one column volume elution buffer (300 mM NaCl, 20 mM HEPES, pH 7.5, 10% (v/v) glycerol, 1 mM DTT, 300 mM imidazole). The eluate was separated by SDS-PAGE. Fractions containing Slx9 were identified by Coomassie staining and Western blot analysis using an anti-His antibody (Santa Cruz, Dallas, TX, USA). Positive fractions were combined, and the buffer was exchanged, by dialysis, to elution buffer without imidazole. The protein was concentrated using a Vivaspin 6 Centrifugal Concentrator (10 kDa cutoff, Sartorius, Goettingen, Germany). The protein concentration was measured by Bradford and by SDS-PAGE using known amounts of bovine serum albumin (BSA) as a standard.

### 4.3. In Vitro Folding and Analysis of G4 Structures and Annealing of Control DNA Structures

The folding of DNA oligodeoxynucleotides into G4 structures was performed as previously described [33]. G4 structure formation was confirmed by 7% SDS-PAGE and circular dichroism (CD) measurements. Oligodeoxynucleotides for control DNA structures were also used as previously published [34] (Appendix A). Annealing was performed in annealing buffer (50 mM HEPES, 2 mM magnesium acetate, 100 mM potassium acetate) for 1 min at 98 °C, 60 min at 37 °C, and 30 min at 22 °C. G4 structures and annealed control DNA structures for binding studies were desalted using illustra MicroSpin G-25 columns (GE Healthcare, Boston, MA, USA).

### 4.4. Binding Studies

DNA (20 pmol) was 5′-labelled with 25 µCi [γ-^32^P] ATP by T4 polynucleotide kinase (NEB, Ipswich, UK). G4 and G4_mut_ structures were purified by 7 % SDS-PAGE. Control DNA (ds, bubble, fork, 4 fork) was purified using illustra MicroSpin G-25 columns (GE Healthcare).

DNA–protein binding was analyzed by double-filter-binding assays [35] using a 96-well Bio-Dot SF apparatus (Bio-Rad, Hercules, CA, USA) and 10 nM DNA in binding buffer (50 mM Tris/HCl pH 8.0, 125 mM KCl, 5 mM DTT, 10% (v/v) glycerol [13]). Protein concentrations increased from 0 to 65 µM Slx9. After incubation on ice for 30 min, the reactions were filtered through a nitrocellulose and a positively charged nylon membrane, followed by three washing steps with binding buffer with no glycerol. The membranes were dried and analyzed by phosphorimaging on a Typhoon FLA 7000 (GE Healthcare). Percentage values of bound Slx9 were determined using ImageQuant, and were used to obtain dissociation equilibrium constants (*K_d_*) by curve fitting using nonlinear regression (GraphPad Prism, San Diego, CA, USA).

### 4.5. ChIP-Seq -and ChIP Plus qPCR Analysis

ChIP experiments were performed as previously described [8,59]. For ChIP-Seq, chromatin was sheared to an average length of 200 bp using a S220 focused ultrasonicator (Covaris, Brighton, UK). For conventional ChIP, the DNA was sheared to an average length of 250 bp using a Branson sonifier W250-D (50% amplitude, 50% duty cycle, 5 × 5 pulses). The applied parameters for Covaris were 140 W, 5% duty, and 200 cycles/burst for 20 min at 4 °C. The pulldown was performed using a c-Myc antibody (Clontech, Kyoto, Japan). For genome-wide sequencing, DNA was treated according to manufacturer’s instructions (NEBNext ChIP-seq Library Prep Master Mix Set for Illumina, NEB) and submitted to deep sequencing (Illumina Nextseq500 sequencer, San Diego, CA, USA). Obtained sequence reads were aligned to the yeast reference genome (sacCer3) with bowtie [60]. After the alignment, the number of reads was normalized to the sample with the lowest number of reads. Binding regions were identified by using the program ‘Model-based Analysis for ChIP-Seq’ (MACS 2.0) with default settings for narrow peaks [61]. Appendix A contains all Slx9 peaks. The ChIP input sample was used as a control. MEME-based motif elicitation was used to identify a consensus motif within the FASTA file from the binding regions identified by MACS 2.0 (Liu lab, Boston, MA, USA) [62]. Overlap of binding sites with other genomic features and binding regions was determined using in house PERL scripts based on a permutation analysis between query and subject features.

For ChIP-qPCR analyses, Slx9 was endogenously Myc-tagged at the C-terminal [63]. Asynchronous cultures were grown to OD (660 nm) of 0.4–0.6 and crosslinked with 1% formaldehyde for 5 min followed by quenching of the crosslinking by the addition of 125 mM glycine. ChIP experiments were carried out as described above, and the immunoprecipitated sample (IP) was subjected to real-time PCR using a SYBR green mix (Roche, Basel, Swiss). Details of the primers used in this study are in Appendix A. Binding was calculated as ratio of IP/Input for the specific regions. Student’s *t*-test was used to calculate the statistical significance.

### 4.6. Growth Assay

The strains used for growth assays are listed in Appendix A. Overnight cultures of *S. cerevisiae* strains were used to inoculate YPD to a starting OD (660 nm) of 0.1. Cultures were grown at 30 °C until OD (660 nm) ≥ 1 was reached. Measurements were taken in 60 min intervals, and doubling times were calculated from log-phase OD (660 nm) values. Phen-DC_3_ (10 µM) or 8 µM *N*-methyl mesoporphyrin IX (NMM) was added to the medium to perform growth assays under G4-stabilizing conditions. Growth curves were performed in triplicates.

### 4.7. Spot Assay

Yeast cultures were inoculated at an OD (660 nm) of 0.15 using a stationary *S. cerevisiae* culture, and grown at 30 °C until OD (660 nm) ≥ 0.8 was reached. All yeast cultures were diluted to OD (660 nm) = 0.8 and a dilution series with six 1:5 dilutions were prepared in a sterile 96-well plate. From each dilution, 3 μL were spotted on a plate, and after drying, incubated at 30 °C. After 2 days, the plates were scanned and growth of strains on different media was compared to estimate the growth defects.

## Figures and Tables

**Figure 1 molecules-24-01774-f001:**
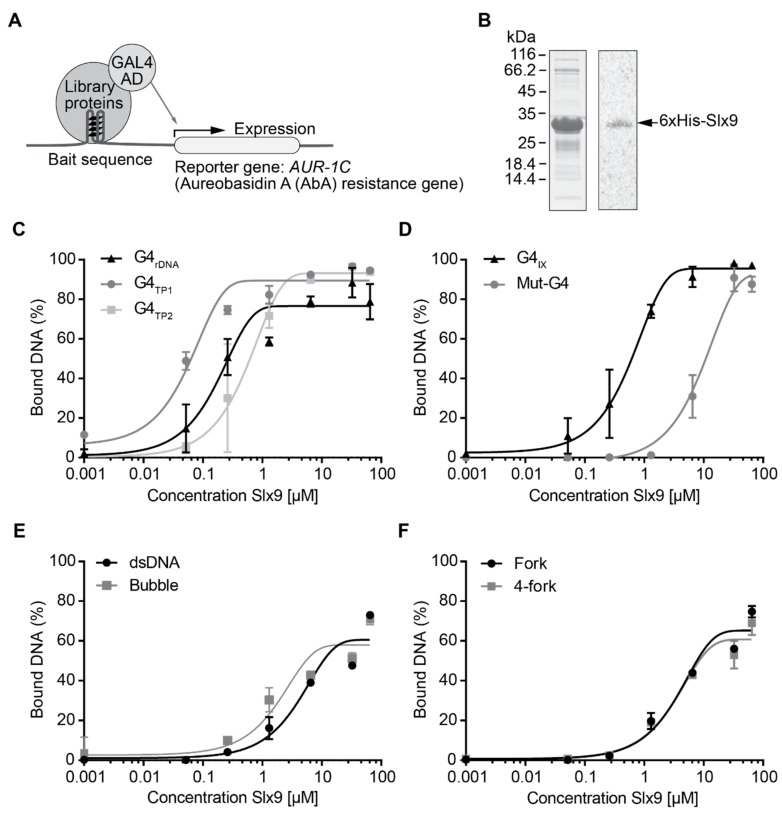
Slx9 is a novel in vitro G4-binding protein. (**A**) Illustration of the yeast one-hybrid screen experimental setup. (**B**) Coomassie staining and Western blot analysis of purified Slx9 protein. The Western blot was performed using an anti-His antibody. 6×His-Slx9 (30 kDa) was detectable between the 25 and 35 kDa marker (arrow). (**C**) Quantification of Slx9 binding to different G4 structures by filter-binding assay, plotted in log scale. Slx9 shows binding to all tested G4 structures with *K_d_* values of 0.55 ± 0.08 µM (G4_IX_), 0.21 ± 0.04 µM (G4_rDNA_), 0.04 ± 0.01 µM (G4_TP1_), and 0.53 ± 0.10 µM (G4_TP2_). (**D**) Slx9 binding to a G4 structure from a selected region of chromosome IX (black) and a mutated version of this G4 motif which cannot fold G4 structures but is 95% identical (grey). (**E**,**F**) Slx9 binding to other DNA structures such as dsDNA, bubble, forked, and 4-fork substrates. Slx9 showed less affinity to the tested control DNA structures: *K_d_* 15.69 ± 3.57 µM (G4_mut_), 5.27 ± 1.18 µM (dsDNA), 1.73 ± 0.42 µM (bubble), 4.21 ± 0.64 µM (fork), 3.72 ± 0.62 µM (4-fork). Plotted results were based on the average of three independent experiments (n = 3).

**Figure 2 molecules-24-01774-f002:**
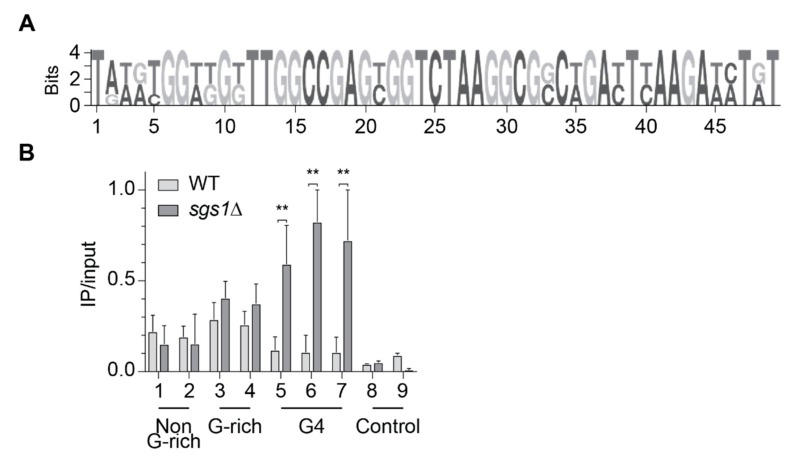
Slx9 binds, genome-wide, to G-rich regions. (**A**) Using Slx9-Myc, two independent ChIP-seq analyses were obtained. Peaks were called using MACS 2.0. A G-rich binding motif was obtained by MEME using the mean of two independent Slx9-Myc ChIP-seq experiments. The binding motif was identified in ~16% of the peaks. (**B**) Slx9 binding was determined to nine endogenous regions (see Appendix A for primer information) by ChIP and qPCR analysis. Selected regions have been identified as binding and non-binding regions in ChIP-seq. Different sequence compositions were chosen for analysis: non-G-rich (binding) and G-rich (binding) sequences, G4 motifs (no binding), and control regions (no binding). G4 motifs contained G-tracts of 3 guanines, resulting in stable G4 structure formation under G4-forming conditions. G-rich sequences could adopt less stable G4 structures, if at all. Binding of Slx9-Myc was monitored in wild type (light) and *sgs1Δ* (dark) cells. Significance was calculated based on Student’s *t*-test (** *p*-value < 0.001). All depicted experiments were performed with at least n ≥ 3 biological replicates.

**Figure 3 molecules-24-01774-f003:**
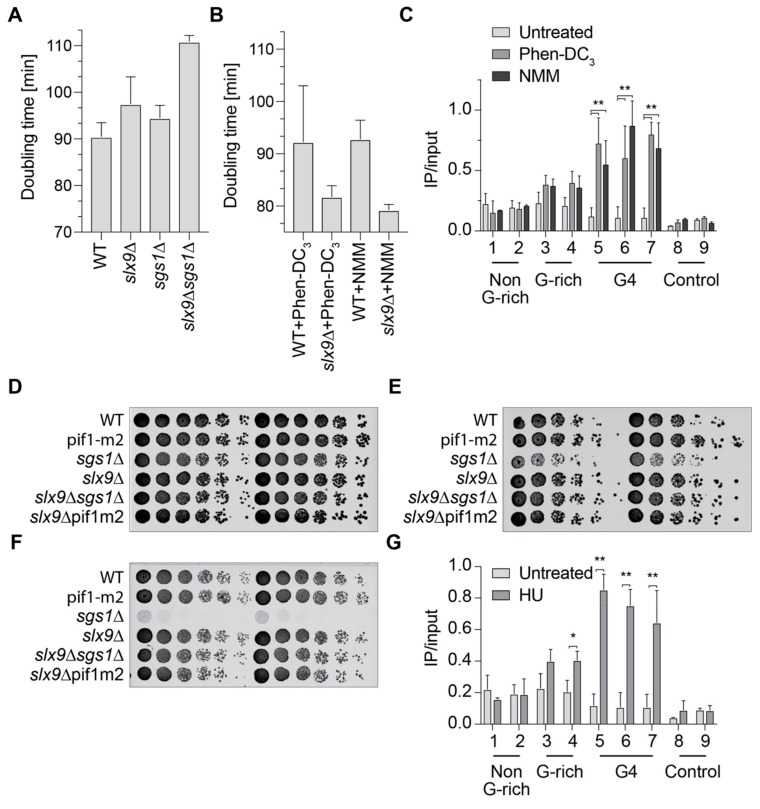
Slx9 supports the recognition of G4 structures in vivo. (**A**) Growth curves were performed and doubling times (minutes) were calculated using the indicated yeast strains. (**B**) Doubling times in the presence of 10 µM Phen-DC_3_ and 8 µM NMM. (**C**) ChIP analysis and qPCR of Slx9 binding in untreated (light) and in the presence of either Phen-DC_3_ (grey) or NMM (dark). (**D**–**F**) Different concentrations of yeast cells were spotted on rich media in a serial dilution to determine growth changes and sensitivity. (**D**) Yeast growth on untreated YPD plates. (**E**) Yeast cells spotted on rich media containing 100 mM hydroxyurea (HU). (**F**) Yeast cells spotted on rich media and incubated with 25 J/m^2^ UV light (254 nm). (**G**) ChIP analysis and qPCR of Slx9 binding in the presence (dark) and absence (light—see also Figure 2B) of 50 mM HU. Significance was calculated based on the Student’s *t*-test (** *p*-value < 0.001, * *p*-value < 0.01). All depicted experiments were performed with at least n ≥ 3 biological replicates.

**Figure 4 molecules-24-01774-f004:**
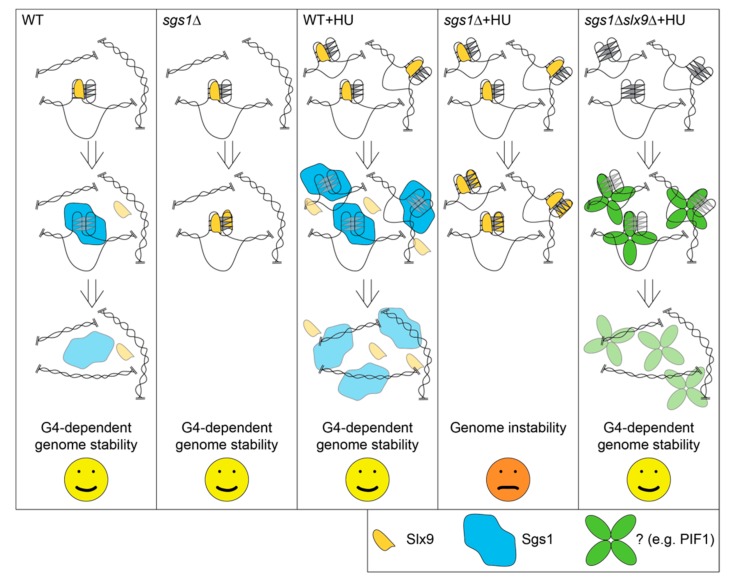
Model detailing how Slx9 recognizes folded G4 structures.

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
