# Peer review of "A Novel G-Quadruplex Binding Protein in Yeast—Slx9"

_molecules, 2019, doi:10.3390/molecules24091774_

Round 1

Reviewer 1 Report

Using a one-hybrid screen, Götz et al identified Slx9 as a novel G4-binding protein and confirmed binding by classical in vitro experiments. The authors performed genome-wide ChIP-seq analysis and found that Slx9 binds only insignificantly to G-rich regions in Saccharomyces cerevisiae in vivo Interestingly, this protein is known to genetically interacts with the yeast RecQ helicase Sgs1; binding to G-rich motif was found to be increased in the absence of Sgs1. They present a model to show how Slx9 recognizes G4 structures. This is an interesting manuscript that deserves to be published after minor revision:

Figure 1C: it is difficult to gauge recognition of various G4 structures as binding is essentially complete at low concentrations. Choose different Y-axis limits? Log-scale?

Figure 2B: The legend could be more precise – the reader may appreciate to understand the difference between G-rich and G4 categories without looking at Sup info. How is “G4” defined for this experiment?

Figure 3AB: Y-axis : units are missing (minutes)

Figure 4: the term “G4-driven genome stability” may be misleading? 

Author Response

We would like to start with thanking both referees for the fast and positive review process. We appreciate that he/she took the time to read our manuscript about Slx9 carefully and made positive and constructive suggestions. Please see below (highlighted in yellow) our responses to your specific concerns.

Reviewer 2 Report

Gotz et al. have identified a G4 quadruplex specific protein Slx1 by yeast one-hybrid approach. They have performed a series of in vitro and in vivo experiments to delineate the role of Slx1 in the context of genome. While Slx1 definitely seems to show affinity for G4 regions in 'in vitro' binding assays, its role on G4 regions in cellular context needs to be further clarified. 

Specific comments:

1. please correct line 52: Typo: lie

2. Fig. 1E-F: Based on their fits, the binding affinity of Slx1 for bubble, ds DNA and fork looks comparable to that for G4 sequences in C and D, even though the binding is not going to 100%. May be if the points up to the first 20 µM protein concentration are shown separately the differences will become more obvious. 

Also, Slx1 seems to show a very similar affinity for fork compared to G4 structures. Would it be possible for the authors to provide some explanation?

3. As wild type yeast contains Sgs1, what is the significance of Slx9 in that context? In Fig 2B, the authors report that in the presence of Sgs1 there is minimal binding of Slx9 to G4 regions. This makes one think that as long as there is functional Sgs1 in the cell, Slx9 will not be able to bind to G-quadruplex regions.

4. In relaxed G4 regions the binding of Slx1 in WT and Sgs1 (deletion) backgrounds is comparable within the range of error, and the extent of enrichment is much less than that of Sgs1(deletion) in consensus G4 regions. How do authors explain this? Does it mean that Sgs1 does not unfold relaxed G4 regions which seems counterintuitive? The explanation provided by the authors does not match the observations for G-rich and G4 regions in Fig 2B.

5. The changes in doubling time reported do not appear significantly different just by looking at the error bars in the Fig 3 (A, B). Even if one assumes they are significant, the observed variations in the doubling time can still result from Slx1 and Sgs1 by independent pathways. The authors can directly test the competition between Slx1 and Sgs1 for G4 quadruplexes in 'in vitro' binding assays.

Likewise, it is difficult to understand why stabilization of G4 by stabilizing chemicals should lead to enhanced growth in the presence of Slx1 deletion. At least, the authors should test the binding affinity of Slx1 to G4 in the presence of Phen-DC3 and NMM in vitro. This will be a nice way to test their hypothesis.

6. The authors should also test if the affinity of Slx1 for G4 is sensitive to HU addition. They should test their hypothesis that Slx1 shows increased enrichment in G4 regions in the genome due to stabilization of G4 structures by HU. This is a rather easy to do experiment in vitro.

Author Response

(The authors gave the same response as above.)

Round 2

Reviewer 2 Report

Earlier comment: Also, Slx1 seems to show a very similar affinity for fork compared to G4 structures. Would it be possible for the authors to provide some explanation? 

Author's response was:

"We disagree with the referee, because the binding affinities clearly show that similar binding affinities are measured for fork, dsDNA and bubble and that all tested G4s have a much better binding affinity. 

To avoid misunderstanding of the binding data, we replottet die data to log scale and included, in the figure legend all obtained Kd values." 

New comment:

Authors can disagree. But, the error bars for the first few critical data points are very large for G4(TP1) and G4(TP2) in Fig 1(C) and for G4(IX) in Fig1(D), which makes the numbers ambiguous.  It does not make sense to me that the error shown with Kd values are comparable for all the G4 constructs even when the actual error bars shown in the figure are so much different for some of them. The authors need to clarify if the repeat experiments were independently fit to obtain the error or the averaged data was fit? If it was latter then the error values do not provide the limits of Kd. This should be addressed. Alternatively, if any experimental limitations caused the error bars to be that large for the first few data points then that should be described to make the interpretation easier for the reader.  

Author Response

We agree with the referee that the error bars for the first few data points are large. The error bars were fit on the average data for three independent experiments. Although the error bars are large for initial concentrations, that could be due to technical error during quantification of the blots as well as technical limitations (pipetting errors) especially for very low concentrations. This is especially the case, because these experiments where pipette by two different people at different days in the lab, which also leads to fluctuations. Nevertheless, the conclusion that SLX9 binds more strongly to G4 structures than non-G4 or mutant G4 structures; does not change. We have added a sentence to the figure legend, that plotted results are fitted curves based on the average of three independent experiments.